# Quantification of hypoxic regions distant from occlusions in cerebral penetrating arteriole trees

Yidan Xue[1‡], Theodosia Georgakopoulou[2‡], Anne-Eva van der Wijk[2], Tamás I. Józsa[1], Ed van Bavel[2‡], Stephen J. Payne[1,3‡]*

1 Institute of Biomedical Engineering, Department of Engineering Science, University of Oxford, Oxford, United Kingdom, 2 Department of Biomedical Engineering and Physics, Amsterdam University Medical Centers, University of Amsterdam, Amsterdam, The Netherlands, 3 Institute of Applied Mechanics, National Taiwan University, Taipei, Taiwan

‡ YX and TG share first authorship on this work. EB and SP are joint senior authors on this work.
* stephenpayne@ntu.edu.tw

**Data Availability Statement:** All relevant data are within the paper and its Supporting Information files.

## Abstract

The microvasculature plays a key role in oxygen transport in the mammalian brain. Despite the close coupling between cerebral vascular geometry and local oxygen demand, recent experiments have reported that microvascular occlusions can lead to unexpected distant tissue hypoxia and infarction. To better understand the spatial correlation between the hypoxic regions and the occlusion sites, we used both *in vivo* experiments and *in silico* simulations to investigate the effects of occlusions in cerebral penetrating arteriole trees on tissue hypoxia. In a rat model of microembolisation, 25 μm microspheres were injected through the carotid artery to occlude penetrating arterioles. In representative models of human cortical columns, the penetrating arterioles were occluded by simulating the transport of microspheres of the same size and the oxygen transport was simulated using a Green's function method. The locations of microspheres and hypoxic regions were segmented, and two novel distance analyses were implemented to study their spatial correlation. The distant hypoxic regions were found to be present in both experiments and simulations, and mainly due to the hypoperfusion in the region downstream of the occlusion site. Furthermore, a reasonable agreement for the spatial correlation between hypoxic regions and occlusion sites is shown between experiments and simulations, which indicates the good applicability of *in silico* models in understanding the response of cerebral blood flow and oxygen transport to microemboli.

## Author summary

The brain function depends on the continuous oxygen supply through the bloodstream inside the microvasculature. Occlusions in the microvascular network will disturb the oxygen delivery in the brain and result in hypoxic tissues that can lead to infarction and cognitive dysfunction. To aid in understanding the formation of hypoxic tissues caused by

**Funding:** This work was partially funded (TG, TIJ, EvB, SJP) by the European Union's Horizon 2020 research and innovation programme, the INSIST project, under grant agreement No 777072. The funders had no role in study design, data collection and analysis, decision to publish, or preparation of the manuscript.

**Competing interests:** The authors have declared that no competing interests exist.

micro-occlusions in the penetrating arteriole trees, we use rodent experiments and simulations of human vascular networks to study the spatial correlations between the hypoxic regions and the occlusion locations. Our results suggest that hypoxic regions can form distally from the occlusion site, which agrees with the previous observations in the rat brain. These distant hypoxic regions are primarily due to the lack of blood flow in the brain tissues downstream of the occlusion. Moreover, a reasonable agreement of the spatial relationship is found between the experiments and the simulations, which indicates the applicability of *in silico* models to study the effects of microemboli on the brain tissue.

## 1. Introduction

Due to the high metabolic demands and its limited capacity for storage, the mammalian brain depends on near constant cerebral blood flow (CBF) to sustain sufficient nutrient and oxygen delivery [1]. Adequate CBF is maintained by the process of cerebral autoregulation (the response of cerebral vessels to blood pressure changes) [2]. Based on local neuronal activation [3], tissue oxygenation is ensured by a compact and complex microvascular network of arterioles and capillaries [4].

The acute effects that follow a sudden CBF reduction, such as loss of consciousness in case of a syncope [5] and irreversible neurological damage in case of a cardiac arrest or acute ischaemic stroke (AIS) [6,7], underline the brain's vulnerability to anoxia. Next to global CBF disruption and large vessel occlusion, cerebral microinfarcts can also lead to detectable brain damage [8]. Such microinfarcts can result from microemboli [9], which can be released into the cerebral circulation under a number of conditions, including atrial fibrillation, unstable plaques [10], or endovascular treatment (the mechanical removal of a thrombus in AIS patients) [11]. However, due to the low resolution of brain imaging techniques, cerebral microinfarcts in humans are mainly discovered only after post-mortem examination.

Evidence of the effects of micro-occlusions on brain tissue thus primarily comes from rodent studies. Such studies have shown that micro-occlusions, formed in response to the intra-arterial injection of microemboli, can lead to multiple regions of ischaemia, hypoxia and infarction [12–17], cognitive dysfunction [18,19], blood brain barrier permeability, astrogliosis and inflammation [20,21]. Using a sophisticated photothrombotic technique which selectively occludes individual penetrating arterioles, Shih et al. not only monitored the formation and progress of a single microinfarct but also assessed the resulting cognitive deficits [22,23]. Such experiments have also confirmed that arterioles have a poor network of anastomoses [24].

Despite these studies, little quantitative information exists on the relationship between micro-occlusion sites, ischaemic territory, and hypoxic area. Given the known vulnerability of penetrating arterioles to obstruction and the distant effects of a single microinfarct on the surrounding tissue [23], it is important to quantify both the extent to which blood vessel architecture affects tissue oxygenation, and how blood vessel architecture and clot location determine the fate of brain tissue viability after blood flow obstruction. A better understanding of brain oxygenation is not only relevant for an improved understanding of the response of the circulation to stimuli but can serve as a starting point for treatment of cerebrovascular diseases [2,25]. For instance, quantitative data on brain tissue damage caused by microemboli could lead to the improvement of medical devices used for post-AIS endovascular treatment.

To this end, numerical models have recently been developed to aid in understanding the structure of cerebral microvasculature [26–35], and the effects of micro-occlusions on blood flow [35–38] and oxygen transport [39,40]. In our previous work, we simulated the oxygen

transport in cerebral capillary cubes using the Green's function method [32] and quantified the tissue hypoxia responding to different level of micro-occlusions [40]. However, the penetrating arterioles were not included in the previous model, which limited the length scale that it was able to represent and thus made it difficult to compare predictions to the microembolisation experiments [17,41] that aim at the penetrating arterioles [42].

In the present study, we thus investigated further the effects of micro-occlusions in penetrating arteriole trees (the vessels between the capillaries and the pial circulation) on tissue hypoxia using both experimental and numerical methods. In our animal experiments, we injected 25 μm (in diameter) polystyrene microspheres through the common carotid artery (CCA) of rat models, which led to tissue ischaemia and hypoxia. Then we reconstructed part of the intervention hemisphere in each animal and segmented the locations of microspheres and hypoxic regions in a 3D coordinate system. In our simulations, we constructed a number of human cortical column networks, which were then occluded by microspheres of the same size, and the Green's function method was applied to simulate oxygen transport. Two novel distance analyses were conducted to investigate the spatial relationship between hypoxic regions and occlusion sites in both experiments and simulations in an identical manner. This enabled direct comparisons to be made between experimental and numerical results and thus to quantify the relationship between the locations of micro-occlusions and hypoxic regions. These results will be valuable in further understanding the response of cerebral tissue to micro-occlusions.

## 2. Methods

### 2.1. Animal experiments

**2.1.1. Ethics statement.** All animal experiments were approved by the ethics committee of the University of Amsterdam, University Medical Center (permit number: DMF321AA). For the required procedures the ARRIVE guidelines and European Union guidelines for the care laboratory animals (Directive 2010/63/EU) were followed.

**2.1.2. Animal surgeries.** Six Wistar rats (n = 6, 50% female, 16–20 weeks old, Charles River) were used. The animals received *ad libitum* food and water and were pair-housed in standard plastic cages with a 12h:12h light-dark cycle. Surgeries were performed exactly as described earlier by Georgakopoulou et al. [41]. Briefly, animals were anesthetized with a mixture of 100% oxygen and isoflurane (Isoflutek 1000 mg/g; Laboratorios Karizoo SA, Barcelona, Spain). After exposure of the left CCA at the bifurcation level, the external carotid artery and occipital artery were temporarily ligated with a surgical suture (size 6.0). To mimic microembolisation, 200 μl of fluorescent microspheres (DiagPoly Custom made Plain Fluorescent Microparticles, Excitation wavelength 656 nm, Emission wavelength 674 nm, Creative Diagnostics, Shirley, NY; 5500 microspheres were injected of 25 μm in diameter) resuspended in a sterile 2% bovine serum albumin solution of phosphate buffered saline (PBS), was injected via the left CCA using an insulin syringe (29 G) as shown in Fig 1A. Microspheres were lodged in the left hemisphere, leaving the right hemisphere as a control, as depicted in Fig 1C.

**2.1.3. Tissue preparation and immunohistochemistry.** Animals were killed 24h post-surgery. Pimonidazole hydrochloride (60 mg/kg; Hypoxyprobe Pacific Blue Kit, HP15-x, Burlington, MA) was used as a marker of hypoxia and lectin (1 mg/kg; DyLight 594 labeled lycopersicon Esculentum tomato, Vector Laboratories, DL-1177, Burlingame, CA) as a marker of ischaemia [17]. Pimonidazole hydrochloride is a probe which binds to cells that have a partial pressure of oxygen ($PO_2$) smaller than 10 mmHg and lectin is a dye which stains perfused blood vessels intravenously. Brain removal, brain pre-processing, and immunohistochemistry of brain sections (50 μm thick) were performed as described previously [17]. Mouse anti-

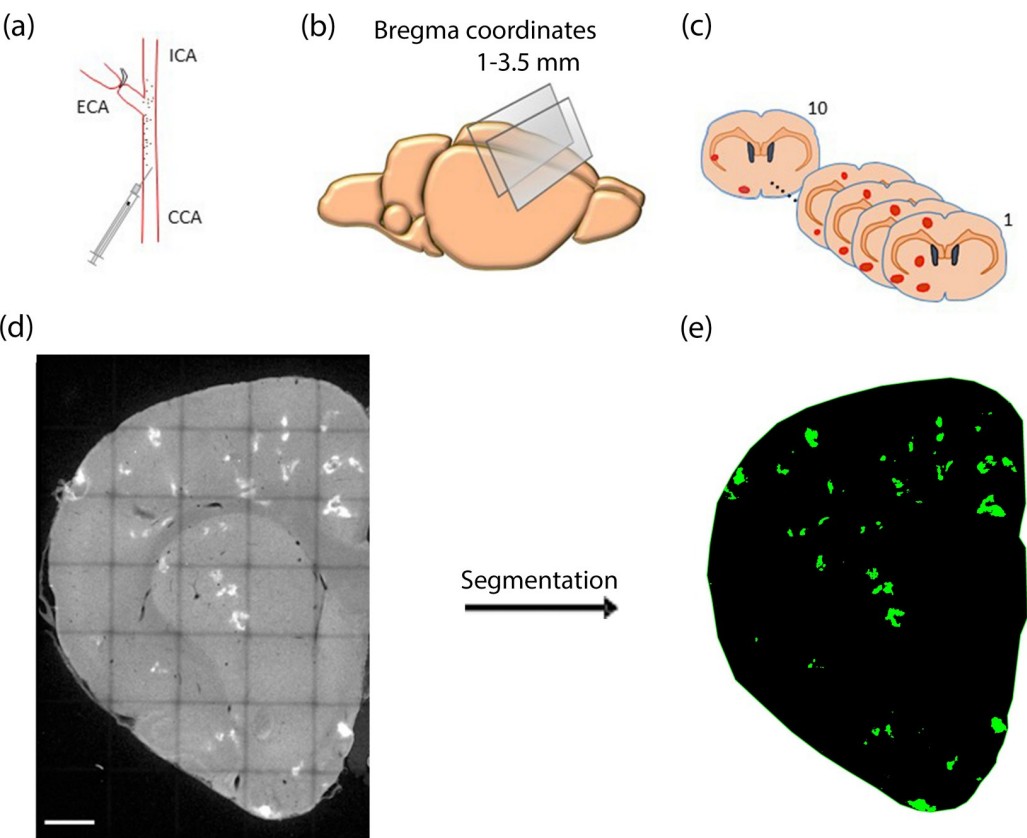

**Fig 1.** (a) Microembolisation model: Injection of microspheres via left common carotid artery (CCA); (b) Brain sections (50 μm thick) were made between 1–3.5 mm bregma coordinates; (c) Ten consecutive coronal brain sections were selected for the brain damage reconstruction. Brain tissue damage (in red) was mainly confined on the intervention hemisphere; (d) Maximum intensity projection (MIP) of a 50 μm thick coronal brain section of the intervention hemisphere with hypoxic regions (white). Tilescan image was made using confocal imaging, Scale bar 1000 μm; (e) The same brain section as in (d) after the segmentation of hypoxic regions (green) in the IMARIS software. CCA—common carotid artery; ICA—internal carotid artery; ECA—external carotid artery.

pimonidazole (Hypoxyprobe Pacific Blue Kit, HP15-x, 1:500) antibody was used to detect Hypoxyprobe and hence to visualize hypoxia.

**2.1.4. 3D reconstruction of intervention hemisphere.** For the spatial analysis between microspheres and brain damage, a 500 μm thick volume of the intervention hemisphere was reconstructed. Ten consecutive coronal brain sections (50 μm thick) devoid of tearing were selected from the forebrain between 1 and 3.5 mm of the bregma as shown in Fig 1B and 1C. Tilescan z-stack images (resolution: x,y: 3.033 μm and z: 5 μm) of the intervention hemisphere were acquired using a confocal laser scanning microscope SP8 (Leica Microsystems, Wetzlar, Germany) with a 10x objective. To facilitate alignment, the z-stack images were converted to maximum intensity projection (MIP) images as depicted in Fig 1D. To this end, the ImageJ software (Rasband, W.S., ImageJ, U. S. National Institutes of Health, Bethesda, Maryland, USA) was utilised, whereas MIP images were aligned using the AMIRA software (Visage Imaging, Inc., San Diego, CA, USA). The resultant z-resolution was thus 50 μm.

**2.1.5. Segmentation of microspheres, ischaemic and hypoxic regions.** Reconstructed brain volumes of the intervention hemisphere were inserted in IMARIS 9.3, a 3D analysis software (Bitplane Inc., St. Paul, MN, USA) and the spot creation wizard and surface creation wizard were used to automatically segment microspheres and hypoxic regions respectively

(Fig 1E). Ischaemic regions were segmented manually by an experienced researcher. The segmented volumes were exported from IMARIS as numbers in excel files and as.tiff binary images, separate for each segmented element (microspheres, ischaemic and hypoxic regions). To quantify how deep in the cortex microspheres were lodged, we manually calculated their distance from the cortical surface, using Leica LAS X software (Leica Microsystems, Germany).

Custom Python codes were used to locate the microspheres and hypoxic regions in the.tiff images to enable distance computations. The 2D binary MIP images were then converted to 3D based on the given z-coordinate of the centre of the coronal brain sections. Images of hypoxic regions were downsampled into $15.165 \times 15.165 \times 50 \ \mu m^3$ voxels to enable direct comparison with the $15 \times 15 \times 15 \ \mu m^3$ resolution employed in the simulations, as described in the following subsection.

### 2.2. Computational models

**2.2.1. Physiologically representative cortical columns.** To simulate the oxygen transport in a cortical column supplied by a single penetrating arteriole, existing models of penetrating arterioles [34] and capillaries [33] were coupled together. Note that these networks were generated from human cerebral microvasculature data [43–45]. To represent a tissue volume of $375 \times 375 \times 1500 \ \mu m^3$, four periodic 375 μm capillary cubes were stacked along the depth of the cortical column. A penetrating arteriole model was then placed at the centre of the column with its inlet aligned with the cortical surface. The branches of the arteriole tree outside the column were trimmed and the boundary nodes were each connected to the closest node of the capillary network. Fig 2 displays typical examples of: (a) a capillary network; (b) a penetrating arteriole tree; and (c) a typical cortical column assembled from these two networks. The geometric information of the 10 example cortical columns generated here is summarised in Table 1.

**2.2.2. Blood flow simulation.** The blood flow in the microvasculature is assumed to be in quasi steady state. The flow is also assumed to be purely laminar because of the low Reynolds

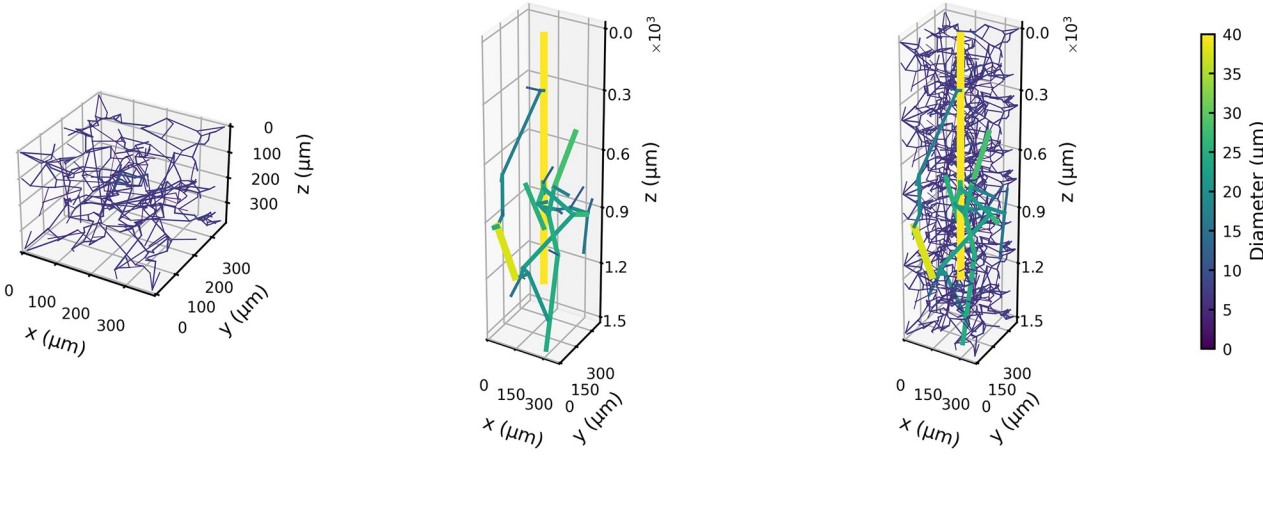

(a) Capillary network          (b) Penetrating arteriole          (c) Cortical column

**Fig 2.** Geometry of (a) a capillary network, (b) a penetrating arteriole and (c) a cortical column built from the penetrating arteriole and the capillary cubes.

**Table 1. Geometric information of cortical columns.**

|  | Simulation | Literature |
|---|---|---|
| Surface density of arterioles | 7.11/mm$^2$ | 5.63-10/mm$^2$ [44,46] |
| Vessel density | 10846±163/mm$^3$ | 10129/mm$^3$ [47] |
| Vascular volume fraction | 2.82±0.22% | 2.70% [47] |

number of about 0.5 in arterioles and about 0.003 in capillaries [48]. In addition, it was assumed that the flow was fully developed and axisymmetric with zero velocity components in the radial and circumferential directions. Hence the Hagen-Poiseuille equation for blood flow in each vessel can be used:

$$\frac{\Delta p}{Q} = \frac{8\mu L}{\pi r^4} \tag{1}$$

where $\Delta p$ is the pressure drop along the vessel, $Q$ is the flow rate, $L$ and $r$ are the vessel length and radius and $\mu$ is the blood viscosity, which is an empirical function of vessel dimeter and haematocrit [49]. The haematocrit was taken to be a constant value of 0.45 [33,40] and the plasma viscosity was taken to be 1.2 mPa·s [34,50].

Note that we assume constant haematocrit here, despite there being multiple studies showing that the haematocrit is not evenly distributed at microvascular bifurcations due to phase separation [51,52]. However, previous studies have shown that the effects of phase separation on blood flow at the length scale modelled here are likely to be second order [53,54]. We also further tested different phase separation models on oxygen transport simulations, the effects of which on tissue oxygenation were found to be of only secondary importance (results not presented here).

Since there is a linear relationship between pressure drop and flow rate in each vessel, the flow field can be solved by using a set of linear equations. Hence an n×n conductance matrix $\Gamma$ can be defined for the network as

$$\Gamma_{ij} = \frac{Q_{ij}}{p_i - p_j} = \frac{\pi r_{ij}^4}{8\mu_{ij}L_{ij}} \tag{2}$$

where n is the total number of vertices of blood vessels in the network and $Q_{ij}$ is the flow rate from $i$ to $j$. $\Gamma_{ij}$ is zero if there is no connection between nodes $i$ and $j$. Then conservation of mass at each node yields

$$\sum_{j=1}^{n} (p_i - p_j)\Gamma_{ij} = S_i \tag{3}$$

where $S_i$ is the source term at node $i$.

By coupling Eqs 2 and 3 and setting the boundary conditions, the pressure at each node and the flow rate in each blood vessel can be solved using a standard linear solver. In this study, all boundary nodes were assumed to have the same pressure, except for the arteriole inlet node at the top of the column. The blood pressure difference between arteriole inlet and boundary nodes was adjusted to maintain a normal perfusion of 55 mL/100mL/min in each column [55]. All blood flow simulations were carried out using custom Python scripts.

**2.2.3. Oxygen transport simulation.** Oxygen transport simulations were performed using the Green's function method [32,56]. This method has recently been implemented in several studies on cerebral oxygen transport [26,31,40]. In the Green's function method, blood vessels are represented by discrete oxygen sources and the tissue region is divided into cuboidal elements which act as oxygen sinks. Inside blood vessels, the blood oxygen concentration

($C_b$) can be represented as a sum of oxygen dissolved in plasma and oxygen carried by haemoglobin as

$$C_b = \alpha_b P_b + C_{Hb} H S \tag{4}$$

where $\alpha_b$ is the blood oxygen solubility, $P_b$ is the blood $PO_2$, $C_{Hb}$ is the oxygen binding capacity per unit volume of red blood cells and $S$ is the oxygen saturation of haemoglobin. This has a non-linear relationship with blood $PO_2$:

$$S = \frac{P_b^N}{P_{50}^N + P_b^N} \tag{5}$$

where $N$ is the Hill equation exponent and $P_{50}$ is the $PO_2$ at half maximal haemoglobin saturation.

The oxygen transport in brain tissue can be simplified to a diffusion-reaction equation:

$$D_t \alpha_t \nabla^2 P_t = M \tag{6}$$

where $D_t$, $\alpha_t$ and $P_t$ are the oxygen diffusion coefficient, the oxygen solubility and $PO_2$ in brain tissue. Metabolic rate of oxygen ($M$) and tissue $PO_2$ are assumed to follow a Michaelis-Menten relationship:

$$M = \frac{M_0 P_t}{P_t + P_0} \tag{7}$$

where $M_0$ is the maximum metabolic rate of oxygen and $P_0$ is the Michaelis constant which is the tissue $PO_2$ when $M$ is half of $M_0$.

According to the potential theory, a Green's function $G(\mathbf{x}, \mathbf{x}^*)$ can be defined between a point source $\mathbf{x}^* = (x_1^*, x_2^*, x_3^*)$ and a point in tissue $\mathbf{x} = (x_1, x_2, x_3)$. In an infinite domain, the solution of the Green's function is

$$G(\mathbf{x}, \mathbf{x}^*) = \frac{1}{4\pi D_t \alpha_t |\mathbf{x} - \mathbf{x}^*|} \tag{8}$$

By integrating the potentials of oxygen from all sources, the $PO_2$ within each tissue element can be solved easily. Full details of the numerical implementation of the Green's function method can be found in [32,56].

The Green's function method was applied to our cortical columns using an open-source C++ package (https://physiology.arizona.edu/people/secomb/greens) developed by Secomb et al. [32]. Custom Python scripts were written to read the network geometry and to post-process the simulation results. The blood $PO_2$ at arteriole inlet was set as 90 mmHg [31]. The tissue region in each column was discretised into voxel of 15×15×15 $\mu m^3$. All model parameters were taken to be the same as used in Secomb et al. [32], except for the maximum metabolic rate of oxygen, which was adjusted to $M_0 = 6.72 \times 10^{-4}$ $cm^3$ oxygen/$cm^3$ tissue/s based on human data [30], to be consistent with our previous numerical study [40].

**2.2.4. Blockage simulation.** To simulate occlusion in the arteriole tree, a bead with a size of 25 $\mu m$ was introduced at the arteriole inlet of each of the 10 cortical columns. The bead was assumed to be carried by the blood flow and to be trapped at the inlet of the first vessel that it reached with a diameter smaller than the bead size of 25 $\mu m$. This led to 6–15 occlusion scenarios in each cortical column, due to different column geometries, which resulted in a total of 91 scenarios in 10 columns. Here we simply assume that the probability of each blockage in a column is the same since there is currently no suitable bead transport model in the cerebral microvasculature. In addition, only one bead is assumed to be present in each arteriole due to

the low bead density found in animal experiments (0.43/column, Section 3.1). Hence, we assume that there are only two situations of the cortical column supplied by one penetrating arteriole: one bead or no bead. After the blockage simulation, the hypoxic regions were identified in each column as the tissue voxels with a $PO_2$ smaller than 10 mmHg, i.e., matching the experimental setting.

## 2.3. Distance calculations for both experimental and simulated data

Two types of distance analysis were performed to establish spatial relationships between vessel blockages and hypoxic regions, namely the pixel-based Gx function and the hypoxic intensity. These two analyses were conducted in identical ways in the *in vivo* experiments and *in silico* simulations, which enables high-fidelity comparisons to be made between the two. Note that the regions outside of the brain tissue in experiments and outside of the cortical columns in simulations were excluded from these distance analyses.

The pixel-based Gx function is defined here as the cumulative fraction of distances from the hypoxic regions to their closest microspheres:

$$\text{Pixel-based Gx } (d) = P(\delta < d) \tag{9}$$

Where $d$ is the distance, P is the probability and $\delta$ is the distance from a hypoxic pixel to its closest microsphere. Note that this analysis is different from the Gx function conducted in our previous study, which used the centre of each hypoxic region instead [41]. In the simulations, the small number of voxels with $PO_2$ below 10 mmHg under healthy condition were excluded from the pixel-based Gx function analysis.

The hypoxic intensity is defined as the volume fraction of hypoxic tissue in a sphere of a certain radius around a microsphere, which is thus a function of distance. Around each microsphere the hypoxic intensity was calculated up to a radius of 500 μm with a step size of 50 μm. In each cortical column, the hypoxic intensity was averaged for all possible blockage sites since we have assumed the same probability of each blockage scenario in each column.

To investigate the spatial correlations between hypoxic regions and microspheres in the experiments, we ran Monte-Carlo simulations of microspheres; these simulations provided the control situation against which we compared the randomness of the distributions obtained here. The control points were randomly generated in the intervention hemisphere in each animal, where the density of points was equal to the *in vivo* bead density in the same brain region. Then the same hypoxic intensity analyses were conducted around these control points, which were compared against the hypoxic intensity calculated in experiments.

## 3. Results

### 3.1. Ischaemic and hypoxic distribution patterns after blocking the cerebral arterioles in a rat model of microembolisation

Table 2 shows the number and density of microspheres and the resulting ischaemia and hypoxia per animal. The density of 25 μm microsphere of 2.03/mm$^3$ (average of 6 animals) results in a volume fraction of $1.66 \times 10^{-5}$ mL beads per mL cerebral tissue. If we assume that the beads are entirely present in a territory of 100 mL supplied by a middle cerebral artery in a human, the total thrombus volume will be $1.66 \times 10^{-3}$ mL. This value is significantly smaller than a typical thrombus volume of 0.17 mL [57], which indicates that the micro-occlusions considered in this study correspond to a case where only about 1% of a larger thrombus is embolised during thrombectomy and thrombolysis.

**Table 2. Number and density of microspheres, total brain volume and % of ischaemia and hypoxia per animal.** These numbers were measured in the reconstructed brain volumes with a thickness of 500 μm.

| Animal | Number of microspheres | Microspheres density (1/mm³) | Total volume of 10 brain sections (mm³) | % of hemisphere being ischaemic | % of hemisphere being hypoxic |
|---|---|---|---|---|---|
| 1 | 25 | 1.23 | 20.32 | 0.51% | 0.81% |
| 2 | 41 | 1.97 | 20.86 | 1.90% | 1.40% |
| 3 | 47 | 1.89 | 24.84 | 2.69% | 2.38% |
| 4 | 26 | 1.29 | 20.18 | 0.64% | 0.30% |
| 5 | 88 | 4.07 | 21.61 | 2.19% | 1.68% |
| 6 | 35 | 1.73 | 20.28 | 0.98% | 1.15% |

A typical example of a blocked arteriole is shown in Fig 3 where a single 25 μm microsphere blocks an arteriole of similar size and leads to ischaemia (absence of lectin staining-red) and hypoxia (green). Since the animals are killed 24h post-surgery, neuronal cell death has already taken place which explains in some cases the absence of hypoxic cells from the lesion core. The implications of this will be discussed later.

Fig 4 shows the distribution of microspheres (white) and the resulting ischaemic (red) and hypoxic (green) regions in each animal. Due to the stochastic nature of the microsphere distribution the resulting ischaemia and hypoxia are not confined to one brain region but are dispersed throughout the brain structures (Cortex, Striatum, Corpus callosum). We found that 73% of the microspheres were lodged in the cortex with the remaining found in deeper brain structures. From the microspheres lodged in the cortex only 0.07% were found on the cortical surface and the majority were lodged at a mean distance of 940.69 ±122.54 μm from the cortical surface. Although ischaemia and hypoxia are distributed heterogeneously, there is a reasonable match (36.1±5.3% of hypoxic regions are also ischaemic) between the two in the affected brain hemisphere within each animal (yellow colour in overlay channel Fig 4).

## 3.2. 3D distance analysis reveals two patterns of hypoxic intensity and a strong correlation between microspheres and hypoxic damage

In a previous study where a mixture of microspheres was injected via the CCA we found a spatial correlation over hundreds of micrometres between microspheres and centroids of hypoxic regions by applying a point pattern 3D distance analysis [41]. However, this experimental setup was not suitable to isolate the effects of a single microsphere or a single size of microspheres on brain tissue oxygenation. Thus, here we only injected one size of microspheres and

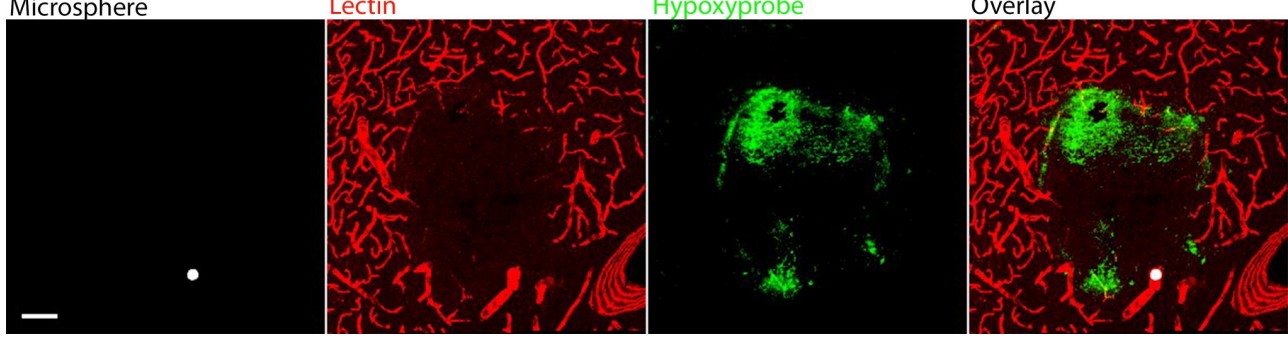

**Fig 3. Maximum intensity projection (MIP) of a 50 μm thick brain section of the intervention hemisphere showing a typical example of a 25 μm (diameter) microsphere causing ischaemia (absence of lectin-red) and hypoxia (green) in the rat brain.** Scale bar 100 μm.

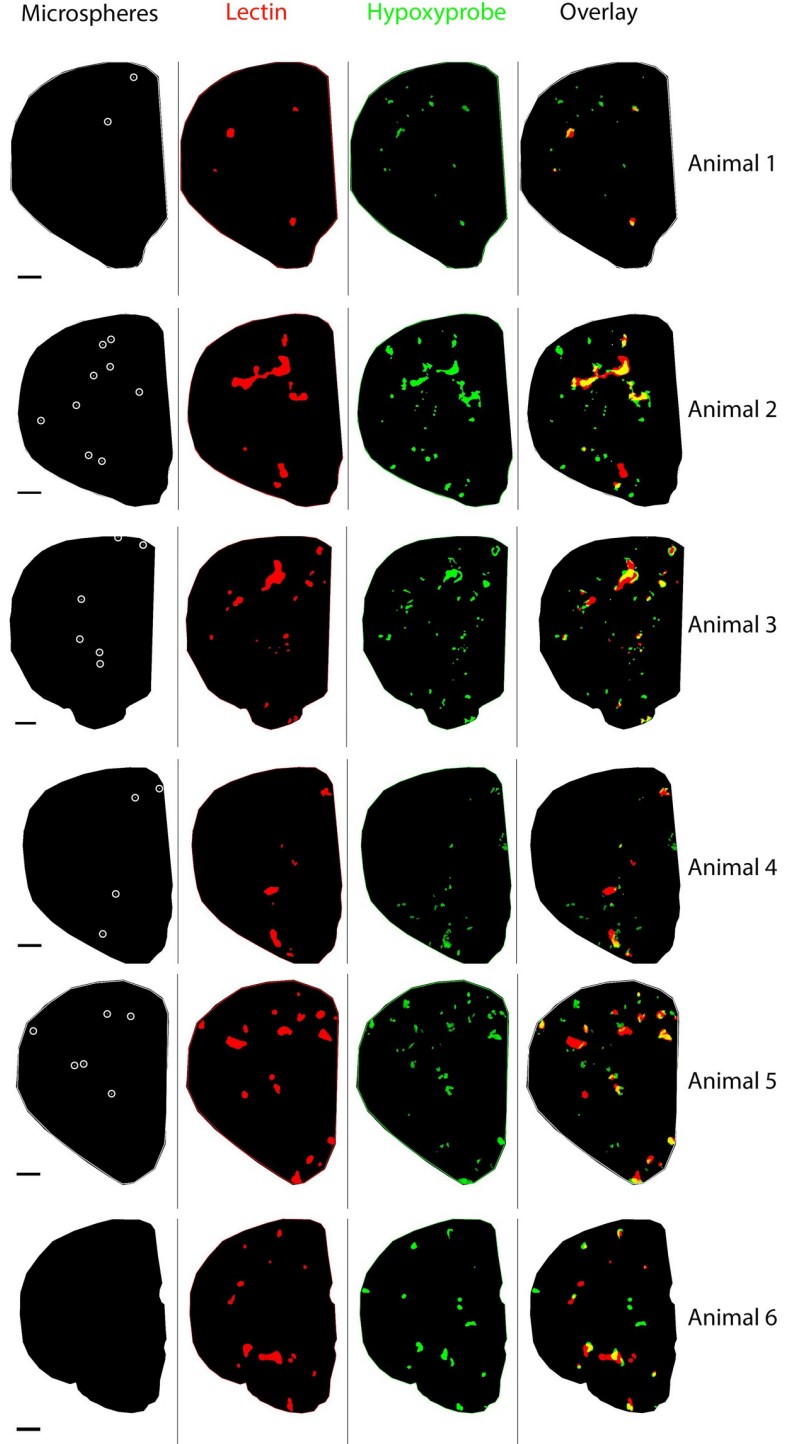

**Fig 4. Per animal a representative segmented brain section of the intervention hemisphere.** From left to right: microspheres (centre of white circles), ischaemia (red), hypoxia (green) and overlay of the three channels. In yellow is shown the overlap between ischaemia and hypoxia. Scale bar 1000 μm.

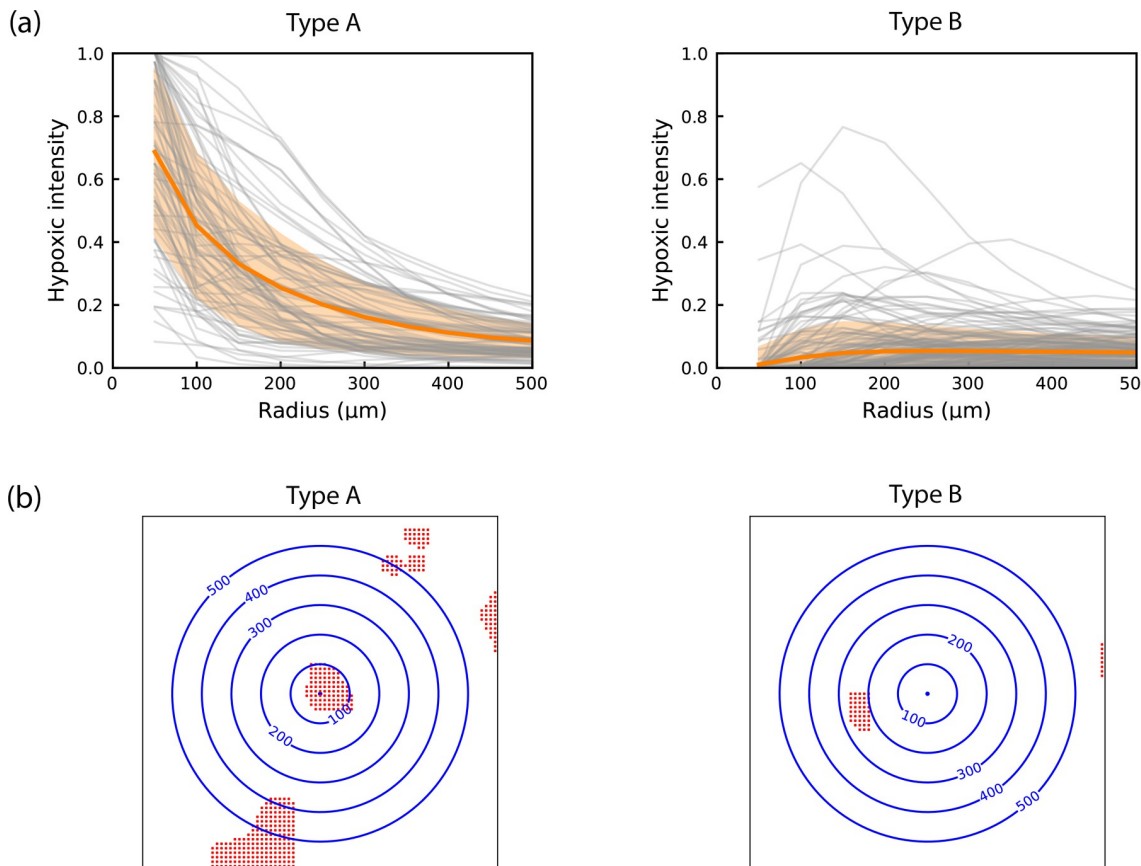

**Fig 5.** (a) Two types of hypoxic intensity. Type A is characterised by negative slope and Type B by positive slope. Mean ± standard deviation (filled areas) (all microspheres of n = 6 animals); (b) Representative examples of the two types of hypoxic intensity. A 2D schematic of the 3D hypoxic intensity analysis. Using a microsphere as a starting point, the number of hypoxic pixels (red) were calculated in spheres of increasing radius (blue concentric circles).

used a different approach for the 3D distance analysis. In particular, we measured the hypoxic intensity as a function of the distance from each microsphere, where each hypoxic intensity curve thus represents the distribution of hypoxic regions around a single 25 μm microsphere.

As shown in S1 Fig, there are cases in all 6 animals where hypoxic intensity decreases to less than half within the first 200 μm and cases where hypoxic intensity starts from zero and reaches 10–20% within the first 200 μm. To elucidate this phenomenon, we categorised the hypoxic intensity patterns into two types (Fig 5A), based on the criterion of whether the slope is negative when the radius increases from 50 to 100 μm (Type A: accounting for 30.9% of the cases when considering all 6 animals), or positive (Type B: 69.1% of the cases). Note that the cases of full hypoxia and no hypoxia inside a radius of 100 μm were categorised as type A and type B respectively. Representative examples for both types are shown in a 2D schematic in Fig 5B.

**Table 3. Percentage of cases of type A and type B hypoxic intensity with or without hypoxic regions within 100 μm of the microsphere.**

|  | Type A | Type B |
|---|---|---|
| Hypoxic regions present within 100 μm of the microsphere (local hypoxia) | 30.9% | 19.8% |
| Hypoxic regions absent within 100 μm of the microsphere (no local hypoxia) | / | 49.2% |

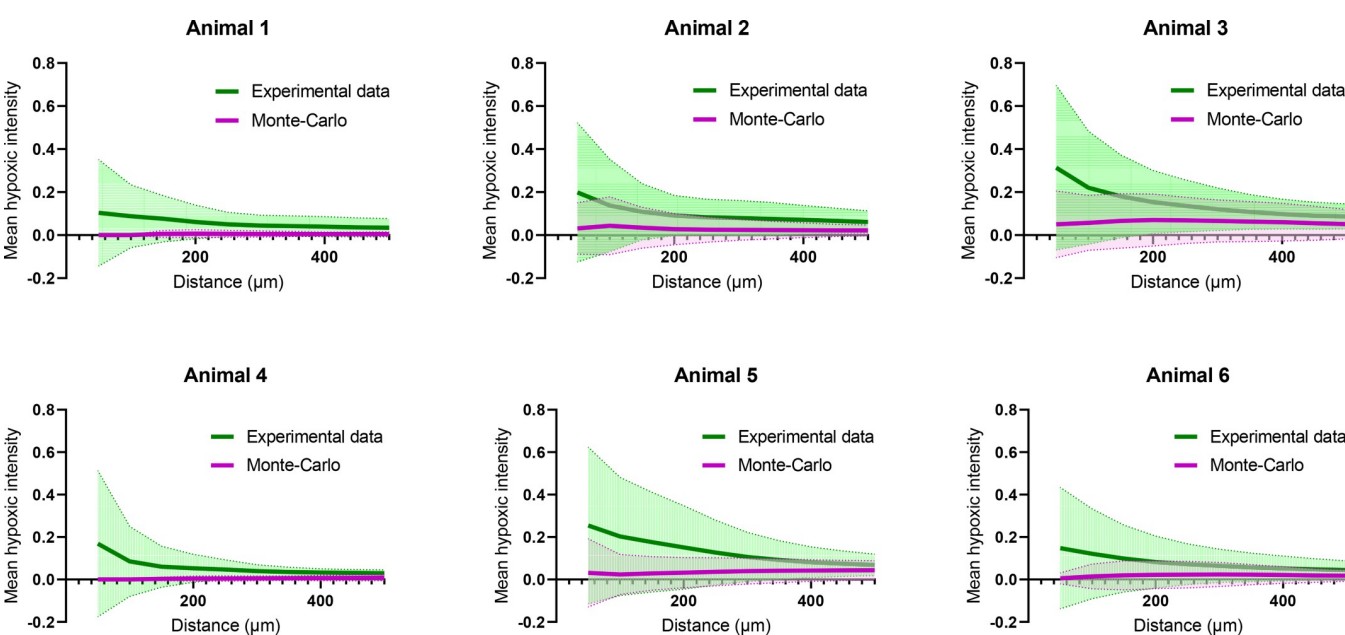

**Fig 6. Per animal experimental data versus Monte-Carlo simulations.** Monte-Carlo simulations: hypoxic intensity as a function of the distance from random control points. Mean ± standard deviation (filled areas) (all microspheres or control points of n = 6 animals).

To further investigate whether the hypoxic regions form locally or distally around the microspheres, we categorised the two types of hypoxic intensity into four subtypes based on the presence or absence of hypoxic regions within 100 μm of the microsphere. The percentage of each subtype in all 6 animals is listed in Table 3. Hypoxic regions are absent within 100 μm of the microsphere in the majority of the type B cases, which also account for 49.2% of all cases. Thus, approximately half of the 25 μm microspheres did not lead to local hypoxia in our experiments.

Next, we tested whether there was a correlation between microspheres and hypoxic pixels. To do this, we ran Monte-Carlo simulations using random control points. The Monte-Carlo simulations shown in Fig 6 for all animals are found to give results that are below the curves of the experimental data: this thus indicates a degree of correlation between individual microspheres and resulting hypoxic regions. Note that in the experimental results we introduced only the hypoxic intensity calculations; the pixel-based Gx function results will be presented in Section 3.5.

It should be noted that these distance calculations do not consider the vessel architecture. Due to tissue deformation and other technical issues such as brain section alignment, blood vessel reconstructions based on confocal imaging of brain sections are not sufficiently accurate. Even in a brain tissue made transparent by means of clearing techniques, blood vessel segmentation remains challenging. This knowledge gap is thus addressed using the numerical simulations presented below.

### 3.3. Heterogeneous distribution of tissue oxygenation in the cortical column

Fig 7 shows the tissue oxygenation in a typical cortical column as solved by the Green's function method. The simulations suggest that oxygen is distributed highly heterogeneously in the tissue within a cortical column. However, it is highly coupled with the microvascular geometry, especially the geometry of the arteriole tree.

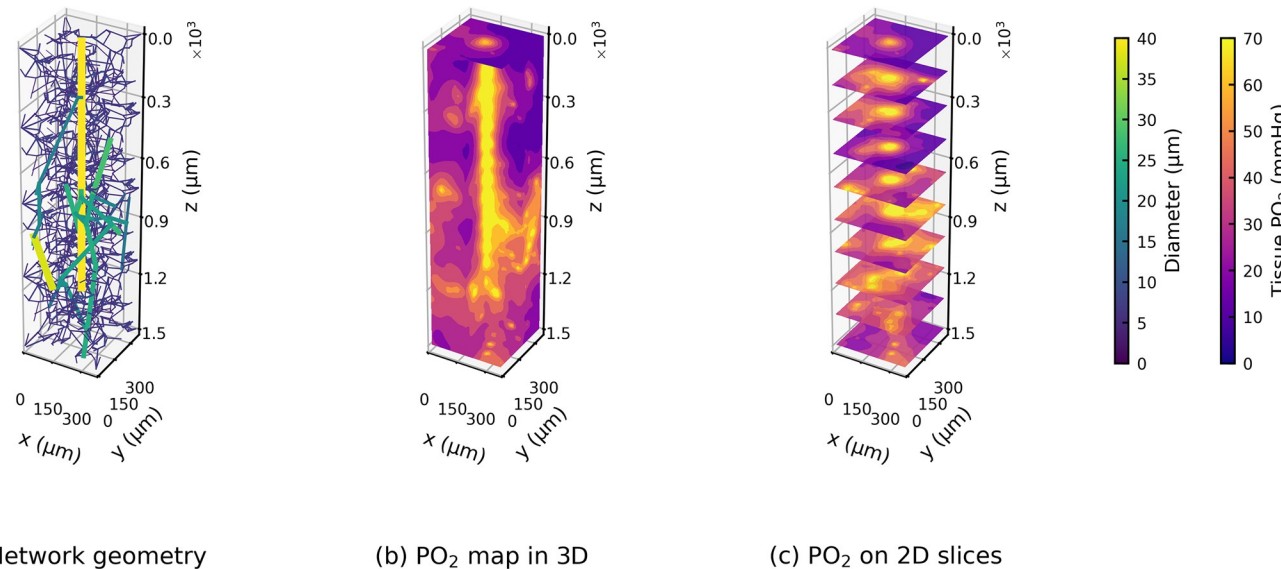

(a) Network geometry (b) PO$_2$ map in 3D (c) PO$_2$ on 2D slices

**Fig 7.** The geometry of a typical cortical column (a) and its tissue oxygenation (b, c) solved by the Green's function method.

To investigate the effects of arteriole and capillary geometries on oxygen transport, we performed two simulations: one in a cortical column with the same arteriole structure as shown in Fig 7A but using a different capillary network (S2 Fig), and the other in a cortical column using the same capillary geometry but connected to a different arteriole tree (S3 Fig). The change in capillary geometry or arteriole geometry leads to a root-mean-square difference of 9.7 or 15.9 mmHg for PO$_2$ in each tissue voxel in the column, respectively. However, these local variations were found to only have negligible effects on the overall PO$_2$ distribution in the column (S4 Fig).

### 3.4. Response of tissue oxygenation to blockage on the arteriole tree

Fig 8 displays three different 25 μm blockage scenarios in the same cortical column, where the blockage locations are indicated by blue spheres. The micro-blockage leads to perfusion drops in the downstream vessels and corresponding regions. The reduction in blood flow (in dark grey) because of the occlusion site is strongly heterogeneous in the column. In addition, the relative perfusion drop in the whole column is 18.5% (a), 50.7% (b) and 8.2% (c) respectively in these three blockage scenarios; this value is thus highly dependent on the blockage location and the specific downstream microvasculature.

The tissue PO$_2$ drop is also distributed unevenly in the column. However, the regions of tissue PO$_2$ drop (in red), which are primarily located around the blood vessels with largest flow rate drops (in dark grey), match those of reduced perfusion closely. The PO$_2$ drop then leads to hypoxic regions (PO$_2$ smaller than 10 mmHg), which are shown as red dots in the bottom row of Fig 8. The tissues near the column centre are found to be unlikely to become hypoxic, due to the oxygen supply from the penetrating arteriole trunk. These simulations thus indicate that the tissue hypoxia caused by occlusion in the arteriole tree is mainly due to the perfusion shortage that occurs in the region downstream of the occlusion site.

### 3.5. Comparisons between simulations and experiments show agreements on pixel-based Gx function and discrepancies on hypoxic intensity

Fig 9 presents the results of two distance analyses in both experiments and simulations. The pixel-based Gx function has a consistent sigmoidal shape in each animal in experiments

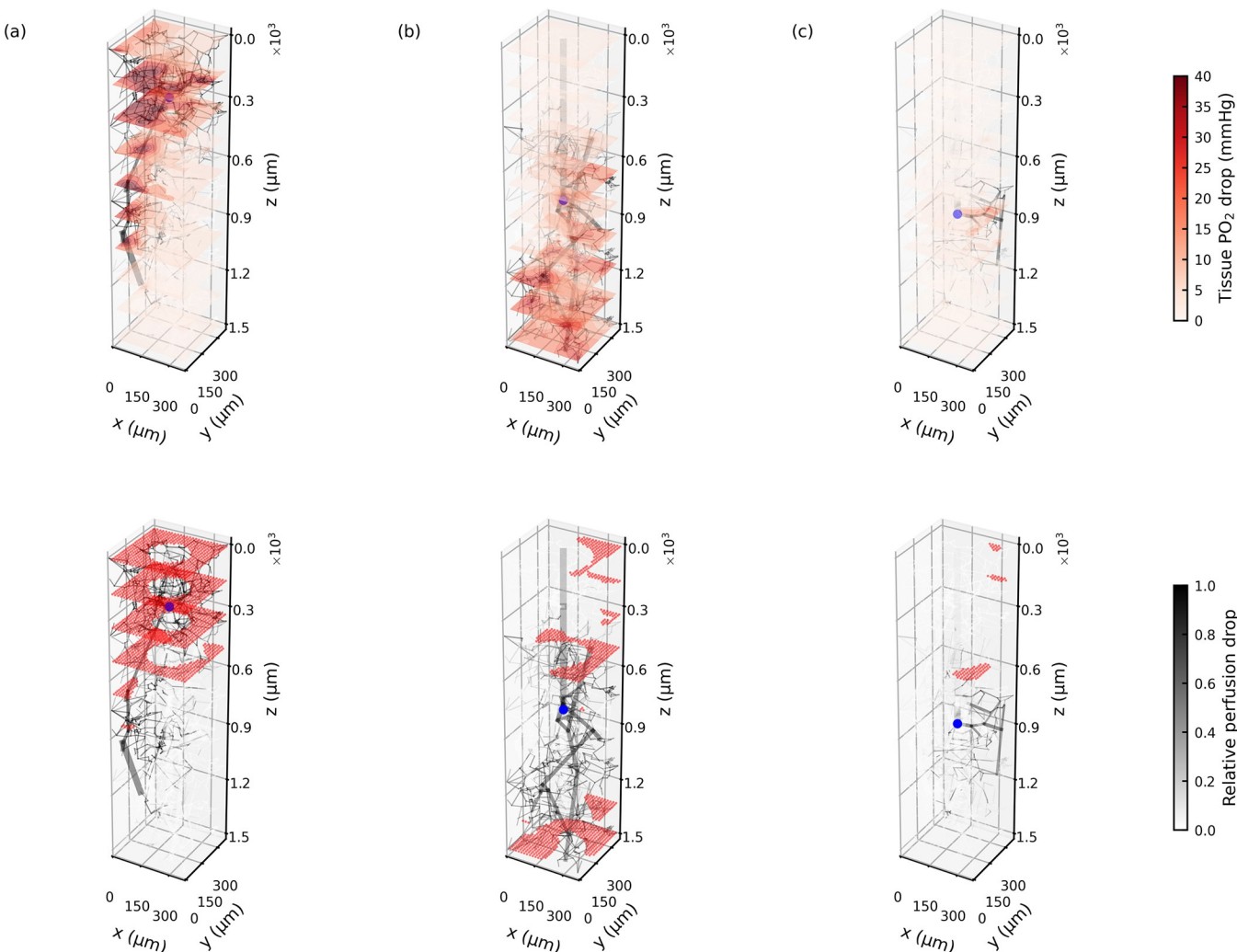

**Fig 8.** The top row shows the drops in relative perfusion and tissue oxygenation in response to 3 different 25 μm blockage (blue sphere) in the same arteriole tree. The bottom row shows the hypoxic regions ($PO_2$ smaller than 10 mmHg, coloured in red) in response to the same blockages.

(Fig 9A) and in each cortical column in simulations (Fig 9B). This results in small standard deviations between each animal and between each cortical column as shown in Fig 9C. In addition, there is a close match for the pixel-based Gx function between experiments and simulations in that they both reach 50% at around 300 μm and reach about 90% at 800 μm. However, the experimental curve is found to start to increase at shorter distances than in the simulations, which indicates that there are more hypoxic regions in the vicinity of occlusion sites in experiments than was found in the simulations.

These discrepancies are more clearly shown in the hypoxic intensity results. The experimental hypoxic intensity curves decrease as the radius increases (Fig 9D), however, most of the simulated hypoxic intensity curves increase at the start (Fig 9E). The reason for this difference is that the experiments have more type A hypoxic intensity (30.9%), while there is only one case out of 91 simulations that is type A (S5 Fig). This leads to some significant differences between experiments and simulations (Fig 9F). The standard deviation of the hypoxic intensity is also larger than that of the pixel-based Gx function in both experiments and simulations. These results will be further discussed in the next section.

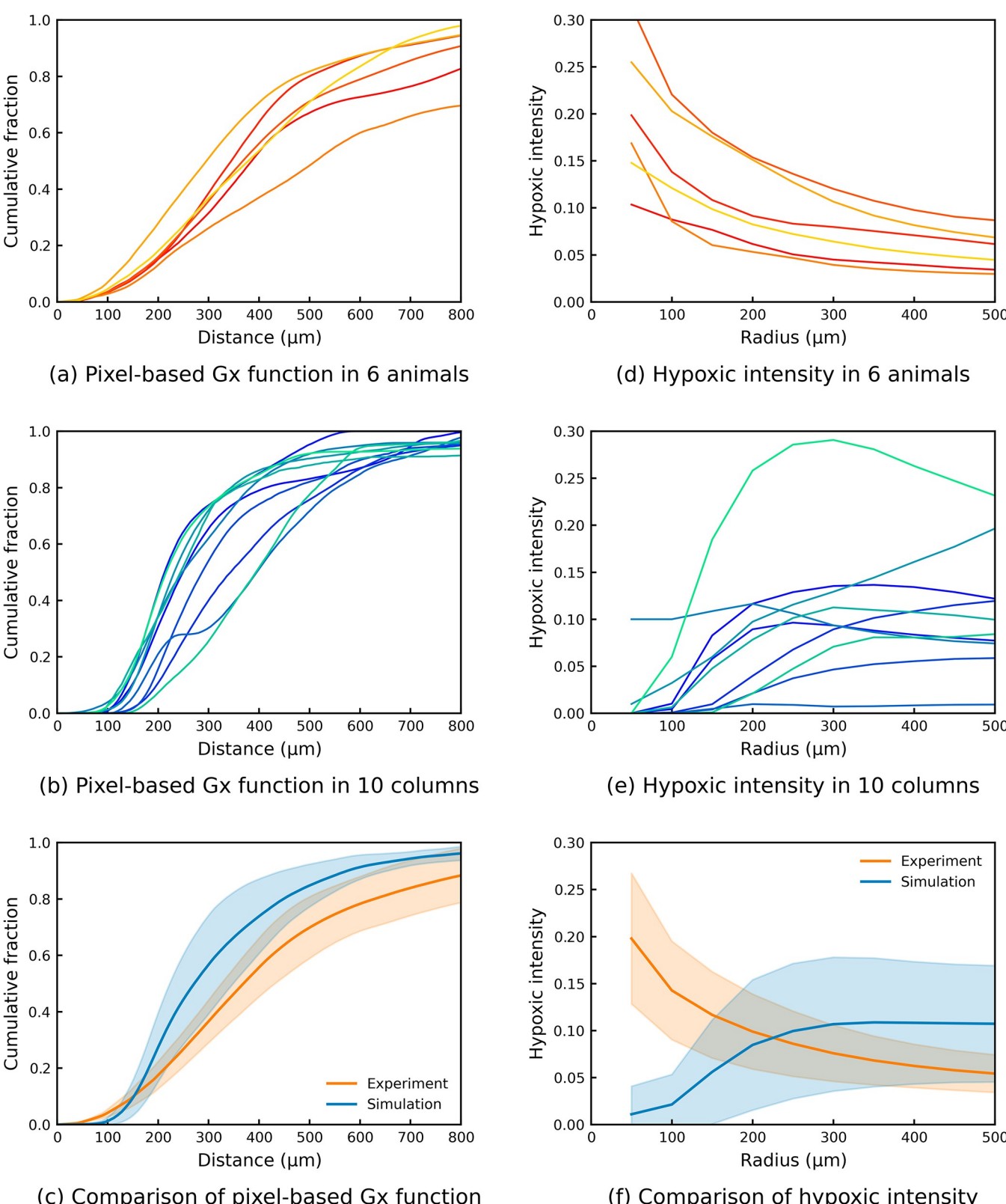

(a) Pixel-based Gx function in 6 animals

(d) Hypoxic intensity in 6 animals

(b) Pixel-based Gx function in 10 columns

(e) Hypoxic intensity in 10 columns

(c) Comparison of pixel-based Gx function

(f) Comparison of hypoxic intensity

**Fig 9. Comparisons of pixel-based Gx function and hypoxic intensity between experiments and simulations.** The error bar of experiments shows the standard deviation between 6 animals. The error bar of simulation shows the standard deviation between 10 cortical columns.

## 4. Discussion

In this study, we examined the effects of micro-occlusions in the cerebral penetrating arteriole trees on tissue hypoxia using both *in vivo* animal experiments and *in silico* simulations. Two novel distance analyses, namely hypoxic intensity and pixel-based Gx function, were carried out identically in both experiments and simulations. The first method focuses on the distribution of hypoxic regions around each microsphere. By comparison, the latter one zooms out to examine the cumulative fraction of distance between the microsphere and the resultant hypoxic regions in the reconstructed or simulated brain tissue volumes. To the best of our knowledge, this is the first study on the spatial relationships between occlusion sites and hypoxic regions that combines both experimental and numerical methods, and the first that considers occlusions over different generations of the penetrating arteriole trees.

Fair agreement in pixel-based Gx function results have been shown between experiments and simulations. These results suggest that hypoxic regions can form distally from the occlusion sites, in agreement with previous experimental findings using mixed microspheres of several sizes [41]. However, other discrepancies in hypoxic intensity have been shown between experiments and simulations. We thus further categorised the hypoxic intensity into two types, where type A decreases and type B increases as the radius increases from 50 to 100 μm. Type A hypoxic intensity represents the scenarios when significant hypoxic regions present near the occlusion site, whereas type B represents the cases when most hypoxic regions form far away from the occlusion site. Type B hypoxic intensity was found to be the dominant case in both experiments (69.1%) and simulations (98.9%). This also agrees with the pixel-based Gx function that more than 90% of the hypoxic regions are more than 150 μm from the microsphere instead of around the occlusion site.

### Reasonable agreement between ischaemic and hypoxic regions

Using a rat model of microembolisation we assessed the percentage of overlap between ischaemia and hypoxia in a limited brain volume of the affected hemisphere, 24h post-surgery. We found ischaemic regions which were not hypoxic, hypoxic regions which were not ischaemic and a 36.1±5.3% overlap between ischaemia and hypoxia. In our previous work where we injected a mixture of microsphere sizes and killed the animals after 1, 3 and 7 days we found that infarction volume at day 7 was similar to that of day 1, suggesting that infarction develops within 24h after microembolisation [17]. As a consequence, we missed in our data the cells which had already undergone cell death. This can explain the ischaemic regions which were not hypoxic. In cases where hypoxic regions did not overlap with ischaemia, we think that larger ischaemic regions, which are formed due to multiple occlusions of the same or different arterial trees, are likely responsible for these results. Hypoxic regions could span beyond the analysed tissue (500 μm thick). As a result, we detect only the hypoxia in our analysed brain tissue, while the ischaemic source is further away in the z-direction. Considering the pathological process of infarct growth and the distal effects of large ischaemic regions, the overlap between ischaemia and hypoxia in our experiments is reasonable.

### Distal hypoxic regions

In the experimental data we found both local (<100 μm, 50.8% of all cases) and distal (>100 μm, 49.2% of all cases) effects due to micro-occlusions. Contrary to previous *in vivo* rodent studies where the occlusion site was highly correlated to brain tissue damage [22–24], we found distal effects after occlusion of penetrating arteriole branches using a rat model of microembolisation [41]. Our simulations suggest that a 25 μm microsphere will occlude a

branch of the penetrating arteriole, which will lead to hypoperfusion downstream of the occlusion site in the corresponding cortical column (Fig 8). The hypoperfusion will then result in distal tissue hypoxia from the occlusion site. This is partially supported by the overlap between ischaemic and hypoxic regions in the experiments.

According to blood vessel diameter measurements, rat penetrating arterioles range in size from 10–30 μm [22]. By injecting microspheres of 25 μm in diameter we should have been able to target the first few branches of penetrating arterioles. We however detected most of the microspheres lodged at a distance > 500 μm below the cortical surface. This discrepancy could be explained by the arterial wall elasticity in combination with the blood flow which may push the microspheres distally from the cortical surface [58]. Since we did not consider the vessel architecture directly, microspheres could block either a penetrating arteriole at a lower level or branches of a penetrating arteriole. Even in the cases where the perfusion volumes may be smaller, the effects of blocked arterioles span a greater distance than was found in *in vivo* occlusions of penetrating arterioles using photothrombosis [22,24].

To understand the discrepancy found between microembolisation and the photothrombotic model we address here some differences between the two techniques. Firstly, in the current experimental study, microspheres are lodged not only in the cortex but also in deep brain structures. Whereas cortical columns are thoroughly examined under controlled circumstances, deeper brain structures and their vessel architecture, perfusion and oxygenation are poorly investigated due to limited depth resolution of live imaging techniques [59]. In addition, despite the high resolution gained by tissue clearing or other modern techniques [60–62], effective image segmentation remains challenging [63]. Secondly, in our microembolisation model the tissue hypoxia can be a result of multiple occluded arteriole(s). In the distance calculations both cases were included, and no distinction was made between the two, since we did not consider blood vessel architecture and could not trace back the effects of every microsphere. Thirdly, we looked at hypoxia 24h after microembolisation, missing the tissue where neuronal cell death has already taken place. Lastly, some microspheres may not necessarily lead to ischaemia or hypoxia, because of a certain degree of collateralization. As a result, microspheres without any detectable brain damage may wrongly be matched to tissue hypoxia created by other occlusion(s). However, we can exclude this scenario based on our simulations, where most single penetrating arteriole occlusions (98.9%) led to distal hypoxic regions thus confirming the experimental findings. Taken together, the inclusion of deep brain structures in our distance analysis and the synergistic effect of multiple occluded arterioles [41] could explain the discrepancies found between the microembolisation versus photothrombotic model, although further detailed work will be required to establish this more accurately.

## Comparisons between simulations and experiments

In this study, numerical simulations were matched with experiments as closely as possible to enable high fidelity comparisons and validations between the two. The validation of *in silico* models against *in vivo* experimental data remains a challenging step for most biomedical simulations, especially for non-linear scenarios like drug delivery and oxygen transport which can result in highly heterogeneous distribution of these substances in tissues. In a recent study, Hartung et al. [39] simulated blood flow and oxygen transport using the reconstructed microvasculature from *in vivo* images, which led to a very good agreement between the simulated oxygen fields and the two-photon oxygen images. This validation [39] focused on tissue oxygenation under healthy scenarios, thus our work provides a further investigation into the pathological effects of micro-occlusions on tissue hypoxia by comparing the spatial relationships.

The validation indicates that the modelling approach is an appropriate one for this scenario and thus supports its wider use in understanding the response of brain tissue to microemboli.

As part of the INSIST project [64], we are developing computational models of AIS at multiple scales to aid in optimising AIS treatments and developing medical devices. The micro-scale models presented here and in previous works [37,40] can in future be coupled with organ-scale models of AIS [65–67] and play a key role in predicting the secondary tissue damage caused by microthrombi after an unsuccessful thrombectomy [68–70].

We next highlight several differences between the experimental and numerical setups. The cerebral microvascular networks used in the simulations [33,34] were generated from the statistical data of the human brain [43–45]. However, the experiments were conducted in rat brains. Despite topological similarities between human and rodent networks, the human capillaries have been found to have longer vessels and larger spacings between vessels than rodent capillaries [71]. In addition, the venules were not included in the simulations, because the occlusions are primarily on the arteriole side. These geometrical differences can potentially lead to discrepancies between simulations and experiments, which should be quantified in future work.

In addition, the blood flow and oxygen transport models in this study are purely steady and passive. However, the hypoxic regions were measured and segmented 24 hours after the microembolisation in experiments. In this time interval, the brain can have active responses to vessel blockage and tissue ischaemia and hypoxia including autoregulation [72,73], pericyte constriction [42,74], microsphere clearance [75,76] and pathological response like infarct formation [17]. These time-dependent events were not considered in the simulations but should be included in future studies.

Discrepancies of hypoxic intensity have been shown between simulations and experiments, which are mainly since there are more type B hypoxic intensity trends found in the simulations than in the experiments. This discrepancy can be partially explained by the differences between the two that we have discussed previously. In the simulations, it is assumed that the bead occludes a branch of the penetrating arteriole with a diameter smaller than 25 μm at the vessel inlet, which tends to be near the column centre. As shown in Fig 8, the hypoxic regions are mainly caused by the hypoperfusion in the regions downstream of the occlusion site and at the column boundary, which will lead to a type B hypoxic intensity. However, in the experiments, microspheres were found to be lodged further away from the cortex surface (mean ± st.dev. = 940.69 ± 122.54 μm), probably due to the vessel wall elasticity in combination with the blood flow. The occlusions will thus more readily happen at the arteriole-capillary transition or the capillary scale. In such a case, more hypoxic regions may form locally (type A) instead of in the regions downstream of the occlusion (type B) in the experiments. It thus leads to fewer type B cases in the experiments than the simulations. In addition, these effects are found to be relatively minor on the pixel-based Gx function, because the hypoxic regions caused by a capillary occlusion tend to be much smaller than these caused by a penetrating arteriole occlusion, which thus only contribute insignificantly to the pixel-based Gx function using the cumulative fraction. However, these hypotheses need further investigations when a fuller description of the vascular geometry is available.

## Limitations

One of the major experimental limitations is the z-resolution of the individual coronal brain sections. Although initially a z-step of 5 μm was used when taking the confocal overview images, the z-resolution changed to 50 μm when converting the images to MIP for the 3D reconstruction of the brain tissue. Due to the low z-resolution and technical issues such as the deformation of brain sections, no blood vessel geometry was considered in the experimental

data. In addition to the limited z-resolution of individual brain sections, the total z-dimension of the reconstructed brain volume (500 μm), was low compared to the x and y dimensions (7000–8000 μm). As a result, the contribution of microspheres outside the reconstructed brain region to tissue hypoxia was hard to predict. This is something that could add uncertainty to the experimental findings. We tackled these limitations by incorporating *in silico* models of cortical columns and comparing the experimental data as closely as possible to the simulations.

One of the steps in our distance analysis workflow involved the segmentation of brain sections to include the brain anatomy in the calculations. Despite gentle handling of the brain tissue during the staining procedure we could not avoid one section of animal 3 becoming broken. For this particular case, we thus used the previous section instead, since the total area and anatomy of consecutive sections were found to be very similar.

One limitation of the simulations is the assumption of one bead or no bead per column, due to the low bead density in the experiments. However, more beads can occlude the same column and the combined effects of multiple blockages on hypoxic regions could affect both the hypoxic intensity and the pixel-based Gx function. Moreover, the occlusions were found to be very mild in the study when we compared the thrombus volume in our models with the thrombus volume measured by clinics. This indicates that there will tend to be more severe micro-occlusions caused by thrombus fragments after an unsuccessful thrombectomy. Hence, we need to be cautious in applying the results presented in this paper directly to clinical studies.

Another limitation is the assumption of the same probability of possible bead locations in the same column since there is currently no available bead transport model in the cerebral microvasculature. This has also been a limitation in our recent studies on the effects of cerebral microthrombi on blood flow and oxygen transport [37,40]. The microemboli transport model will thus need to be developed and coupled with current oxygen transport models in future work.

## Supporting information

**S1 Fig. Per animal the mean hypoxic intensity as a function of the distance from a microsphere.**
(TIF)

**S2 Fig.** The geometry of a column with the same arteriole geometry as shown in Fig 7A but different capillary geometry (a) and its tissue oxygenation (b, c).
(TIF)

**S3 Fig.** The geometry of a column with the same capillary geometry as shown in Fig 7A but different arteriole geometry (a) and its tissue oxygenation (b, c).
(TIF)

**S4 Fig.** Tissue PO2 distribution in cortical columns shown in Figs 7 and S2 (using a different capillary cube) and S3 (using a different penetrating arteriole tree).
(TIF)

**S5 Fig. Two types of hypoxic intensity in simulations.**
(TIF)

**S1 Data. Manuscript data of pixel-based Gx function and hypoxic intensity in simulations and experiments.**
(XLSX)

## Author Contributions

**Conceptualization:** Yidan Xue, Theodosia Georgakopoulou, Tamás I. Józsa, Ed van Bavel, Stephen J. Payne.

**Data curation:** Yidan Xue, Theodosia Georgakopoulou.

**Formal analysis:** Yidan Xue, Theodosia Georgakopoulou.

**Funding acquisition:** Ed van Bavel, Stephen J. Payne.

**Investigation:** Yidan Xue, Theodosia Georgakopoulou, Anne-Eva van der Wijk.

**Methodology:** Yidan Xue, Theodosia Georgakopoulou, Anne-Eva van der Wijk, Tamás I. Józsa.

**Project administration:** Ed van Bavel, Stephen J. Payne.

**Resources:** Ed van Bavel, Stephen J. Payne.

**Software:** Yidan Xue, Theodosia Georgakopoulou.

**Supervision:** Ed van Bavel, Stephen J. Payne.

**Validation:** Yidan Xue, Theodosia Georgakopoulou.

**Visualization:** Yidan Xue, Theodosia Georgakopoulou.

**Writing – original draft:** Yidan Xue, Theodosia Georgakopoulou.

**Writing – review & editing:** Yidan Xue, Theodosia Georgakopoulou, Tamás I. Józsa, Ed van Bavel, Stephen J. Payne.

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
