## [Decision Letter · Decision Letter 0]

1 Jun 2022

Dear Professor Payne,

Thank you very much for submitting your manuscript "Quantification of hypoxic regions distant from occlusions in cerebral penetrating arteriole trees" for consideration at PLOS Computational Biology. As with all papers reviewed by the journal, your manuscript was reviewed by members of the editorial board and by several independent reviewers. The reviewers appreciated the attention to an important topic. Based on the reviews, we are likely to accept this manuscript for publication, providing that you modify the manuscript according to the review recommendations.

Sincerely,

Daniel A Beard

Deputy Editor

PLOS Computational Biology

Daniel Beard

Deputy Editor

PLOS Computational Biology

[LINK]

Reviewer's Responses to Questions

**Comments to the Authors:**

Reviewer #1: The authors present experimental and theoretical results concerning the effect of local occlusion in brain cortex microvasculature on the occurrence of hypoxic regions. This is a medically important topic, because of the damage to brain tissue that can result from microemboli, such as result from breaking up larger thrombi. The results are analyzed in terms of the distribution of distances between occlusion sites and the resulting hypoxic sites. The results indicate some similarities and some differences between the theoretical simulations and the experimental results. The reasons for the discrepancies are not entirely clear, but different distributions of blockage sites through the cortical thickness appear to be a factor. Both the experimental and theoretical aspects of the work are carefully done and clearly presented. Overall, this is a novel approach to an important topic, and opens up new directions for future work on the relationship between local flow blockages and tissue hypoxia/ischemia. Consequently, this is a valuable contribution, even though the work does not lead to definitive conclusions on this topic.

Specific comments

1. Line 321. “pixel-based Gx function” The significance of “Gx” is not explained. This term could be removed without causing any ambiguity.

2. Line 321. It would be helpful to add a comment about the purpose of the “distance analysis,” e.g. to establish spatial relationships between vessel blockages and hypoxic regions.

3. Line 354: should be “number and density of microspheres”

4. Figure 4. The locations of the microspheres are virtually invisible. I suggest adding arrows or circles to show where they are.

5. Line 530. “By comparison, the latter one zooms out…” In principle, the two measures are giving the same information. If f(r) is the hypoxic intensity, then the cumulative distribution is the normalized value of integral(0 to r) (f(r) r^2 dr), which is approximately the pixel-based Gx function.

6. Line 537. “Excellent agreement” is an overstatement. If there was excellent agreement, then the “hypoxic intensity” results could not differ by so much (see previous comment). I would say “fair agreement.”

7. Line 878: Reference seems to be incomplete.

Reviewer #2: This work uses animal experiments and computational modeling to improve understanding of how embolisms in brain microvessels create both ischemia and hypoxia, and in particular how the resulting hypoxia can be either co-localized with the embolism or relatively distant from it. I feel the work is of high quality and novel, and therefore generally suitable for PLOS Computational Biology. I do have several comments I feel the authors should consider.

1. Although the meaning of the hypoxia-embolism distance measure Gx is fairly clear, an exact definition is not given. I feel this would be helpful in interpreting results presented (e.g., delving deeper into apparent similarities between model and experiment in Fig.9a-c), and for comparisons to other work in the future. This could be given in the main part of the manuscript or in Supporting Information.

2. The hypoxia intensity measure used gives the volume fraction of hypoxia in spheres of increasing radii around an embolism (in this case, 25micron sphere). This provides very useful information, but did the authors consider using a differential measure instead, where at each radius the hypoxic density in the corresponding spherical shell is given? This would seem to have potential to give more details on the average spatial distribution of hypoxia, and possibly give a more sensitive comparison between experiment and modeling.

3. It is somewhat surprising how dominated by diffusion from arterioles the calculated tissue PO2 distributions are. Has this effect been verified by comparisons to brain PO2 measurements, and has the intravascular resistance to O2 transport inside arterioles been considered in the model?

4. Can the authors explain the fact that although only 31% of experimental measurements give type A hypoxic behavior (localized near the embolism), the overall effect (Fig.9d) is dominated by type A? (Note that the Fig.9 caption is not fully correct.). This seems to be a relative magnitude effect, so perhaps some other type of averaging could be considered.

5. It is very interesting that in simulations almost 99% of the hypoxic behavior is type B (vs 69% experimentally). It seems the model could easily be used to test the idea that blockages close to a large vessel (ie penetrating arteriole) give type B behavior by using microspheres of decreasing size (20, 15, 10 microns), and looking for a B to A transition. I definitely suggest this approach as it would also address the possibility that 25micron spheres are blocking slightly smaller vessels experimentally.

6. A further test of ideas in the previous comment that the authors could consider is determining the actual size of the vessels blocked experimentally, and also the distance of each microsphere from the nearest penetrating arteriole. Even one of these pieces of information, which I think should be possible to extract from the existing dataset, would help to explain the A/B differences shown in Fig.9, and improve understanding of how and where micro-embolisms form.

**Have the authors made all data and (if applicable) computational code underlying the findings in their manuscript fully available?**

Reviewer #1: Yes

Reviewer #2: **No: **The authors state that data and code will be provided upon reasonable request, but this does not seem to satisfy the PLOS Data policy.

PLOS authors have the option to publish the peer review history of their article (what does this mean?). If published, this will include your full peer review and any attached files.

Reviewer #1: No

Reviewer #2: No

Figure Files:

Data Requirements:

Reproducibility:

References:

---

## [Decision Letter · Decision Letter 1]

14 Jul 2022

Dear Professor Payne,

We are pleased to inform you that your manuscript 'Quantification of hypoxic regions distant from occlusions in cerebral penetrating arteriole trees' has been provisionally accepted for publication in PLOS Computational Biology.

Best regards,

Daniel A Beard

Deputy Editor

PLOS Computational Biology

Daniel Beard

Deputy Editor

PLOS Computational Biology

Reviewer's Responses to Questions

**Comments to the Authors:**

Reviewer #1: The authors have responded appropriately to my previous review.

Reviewer #2: I thank the authors for addressing all of my comments on the original manuscript.

**Have the authors made all data and (if applicable) computational code underlying the findings in their manuscript fully available?**

Reviewer #1: Yes

Reviewer #2: None

PLOS authors have the option to publish the peer review history of their article (what does this mean?). If published, this will include your full peer review and any attached files.

Reviewer #1: No

Reviewer #2: No

---

## [Editor Report · Acceptance letter]

2 Aug 2022

PCOMPBIOL-D-22-00696R1 

Quantification of hypoxic regions distant from occlusions in cerebral penetrating arteriole trees

Dear Dr Payne,

I am pleased to inform you that your manuscript has been formally accepted for publication in PLOS Computational Biology. Your manuscript is now with our production department and you will be notified of the publication date in due course.

With kind regards,

Zsuzsanna Gémesi
